

# Reaching for upper bound ROUGE score of extractive summarization methods

Iskander Akhmetov[1,2], Rustam Mussabayev[2] and Alexander Gelbukh[3]

[1] Kazakh-British Technical University, Almaty, Almaty, Kazakhstan
[2] Institute of Information and Computational Technologies, Almaty, Almaty, Kazakhstan
[3] Instituto Politecnico Nacional, Mexico, Mexico

## ABSTRACT

The extractive text summarization (ETS) method for finding the salient information from a text automatically uses the exact sentences from the source text. In this article, we answer the question of what quality of a summary we can achieve with ETS methods? To maximize the ROUGE-1 score, we used five approaches: (1) adapted reduced variable neighborhood search (RVNS), (2) Greedy algorithm, (3) VNS initialized by Greedy algorithm results, (4) genetic algorithm, and (5) genetic algorithm initialized by the Greedy algorithm results. Furthermore, we ran experiments on articles from the arXive dataset. As a result, we found 0.59 and 0.25 scores for ROUGE-1 and ROUGE-2, respectively achievable by the approach, where the genetic algorithm initialized by the Greedy algorithm results, which happens to yield the best results out of the tested approaches. Moreover, those scores appear to be higher than scores obtained by the current state-of-the-art text summarization models: the best score in the literature for ROUGE-1 on the same data set is 0.46. Therefore, we have room for the development of ETS methods, which are now undeservedly forgotten.

## INTRODUCTION

Automatic text summarization (ATS) is a process of generating a relatively small-sized text out of a bigger one while preserving all the critical information. The research on the problem started in 1958 (*Luhn, 1958*) and saw a huge development in terms of methods, approaches, and applications. The most numerous advancements in the ATS happened after 2003 (*Parker et al., 2011*) when the large data sets and powerful computational resources became available to researchers.

Generally, ATS methods can be classified on the type of input (multi-/single-document), output (extractive/abstractive) and content (informative/indicative); see Fig. 1.

The methods shown in Fig. 1 described as follows:

1. *Input*

(a) Single-document summarization is when we summarize one single document, using only the textual information within and no additional sources.

Corresponding author
Alexander Gelbukh,
gelbukh@gelbukh.com

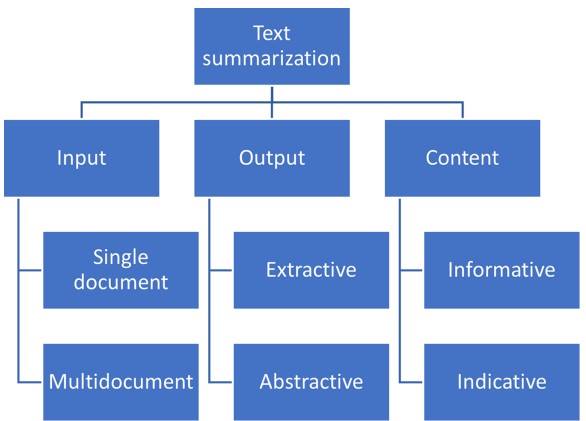

**Figure 1 Classification Automatic Text Summarization methods (*Radev, Hovy & McKeown, 2002*; *Abualigah et al., 2020*).**

(b) Multi-document summarization produces a summary of a set of documents related to a common subject but varying by the time of appearance, size, and source. Applications of the method cover many areas, including literature review in scientific research, business intelligence, government reports, and legal document processing.

2. *Output*

(a) Extractive summary contains only original sentences from the source text, without any change or recombination. Such summaries often lack cohesion between consequent sentences as they are extracted from different parts of the text, taking into account solely the statistical significance of the words they contain.

(b) Abstractive summary is a completely new text generated relying on the information in the source text put through the prism of the opinion and understanding of the information consumed by the reporter. The method requires more sophisticated natural language generation (NLG) models and approaches than extractive methods.

3. *Content*

(a) Informative summaries contain all the critical information from the source text and avoid redundancy. Generally, it is achievable at the 20% compression rate (*Kupiec & Pedersen, 1995*).

(b) Indicative summaries aim at teasing the reader to consume the whole article to stimulate the article purchase or spend time on a long read.

Thus, extractive summarization methods "extract" sentences or other text items, such as words or paragraphs, from the original text to make summaries without making up even a single word. The advantage of these methods is that they are always factually correct according to the processed text. On the other hand, abstractive summarization methods often give related information from sources other than the original text.

The challenging question we want to answer in this article is whether we have room for developing extractive text summarization (ETS) methods. Or are they outdated and have

to be replaced by abstractive text summarization methods? Additionally, we question what maximum summary quality we can achieve using ETS methods.

In this article, to assess the quality of generated summaries, we use ROUGE-1 and ROUGE-2 scoring, which are the quantitative evaluations of the number of words shared by a candidate summary with the reference (or "golden") summary, divided by the number of words in these summaries, and the harmonic mean between these two numbers; see "Evaluation".

Therefore, we define the ATS optimization problem as finding the ultimate set of sentences for the summary to yield the maximum ROUGE score possible. However, the problem belongs to NP-full class of problems, and solving it with the Brute Force algorithm would not be feasible, and we need to find a better way by applying a heuristic algorithm.

For this purpose we compare the use of the variable neighborhood search (VNS) (*Hansen & Mladenović, 2018*; *Hansen et al., 2010*) method; see "Variable Neighborhood Search (VNS)", with a greedy algorithm, which extracts sentences from the source text containing the maximum number of words from the "golden" summary; see "Greedy Algorithm", and finally, with the genetic algorithm.

We also run experiments with variable neighborhood search (VNS) and genetic algorithms initialized by the Greedy solution; see "VNS Initialized by the Greedy" and "Genetic Algorithm Initialized by the Greedy".

The contribution of our research to the scientific knowledge is in (1) discovery of the ETS methods ROUGE score upper bound, (2) a dataset of scientific texts with high-ROUGE score extractive summaries produced by the algorithms discussed in this article, and valuable text statistics (https://data.mendeley.com/datasets/nvsxfcbzdk/1), (3) code to replicate the implemented research (https://github.com/iskander-akhmetov/Reaching-for-Upper-Bound-ROUGE-Score-of-Extractive-Summarization-Methods).

At the same time, we raise a discussion on several important topics for further research in "Discussion".

In "Related Work", we gave a short overview of the research and developments made in the area of ATS. Then, in "Methods and Data" we describe the data used for our experiments and the methods and the Experiment setup is described in "Experiments". In "Results", we show the obtained results, followed by discussion of the issues and thoughts we found during our research in "Discussion", and concluding the work in "Conclusion" with setting out prospects for future work.

## RELATED WORK

Most automatic text summarization (ATS) research papers are devoted to summarization methods. However, few papers research the upper bound of quality achievable by the summaries generated.

*Ceylan et al. (2010)*, working on the texts in the domains of scientific, legal, and news texts, used an exhaustive search strategy to explore the summary space of each domain and found respective probability density function (PDF) of the ROUGE score distributions.

Then using the obtained PDF function, they ranked the summarization systems that existed for the time by percentiles.

Further, *Verma & Lee (2017)* explored the upper bound limits for single and multi-document summary quality on DUC01/02 datasets. However, they made it only for the recall part of the ROUGE scoring metrics, stating that the upper limit for the recall is achieved by using the whole source text as a summary leading to that metric going up as far as 90–100%. Nevertheless, using the entire text as a summary is not what we are looking for in the ATS task.

Abstractive summaries composed by humans using their own words leave little chance for extractive summarization to get a high ROUGE score. *Wang et al. (2017)* propose nine heuristic methods for generating high-quality sentence-based summaries for long texts from five different corpora. They demonstrated that the results achieved by their heuristics methods are close to those of Exhaustive (or Brute Force) algorithms but work much faster (*Wang et al., 2017*).

In this work, we used the VNS heuristic algorithm (*Hansen & Mladenović, 2001*) for finding the set of sentences in the original text to assemble the best ROUGE score summary. VNS iteratively changes the initial random solution and updates the rate of change if no improvement occurs, fixing the best result.

We also applied a Greedy algorithm (*Black, 2005*), widely applied in different text summarization approaches:

- Maximal marginal relevance (MMR) *Carbonell & Goldstein (1998)* struggles to increase relevance while reducing redundancy of the selected sentences.
- Integer linear programming (ILP) *Gillick et al. (2009)*, identifying the key concepts in the summarized text and then greedily selecting the sentences covering those concepts at maximum.
- Submodular selection: optimized semantic graph submodule extraction, built on the text being summarized (*Lin, Bilmes & Xie, 2009*).

Nevertheless, in this article, we use the Greedy algorithm to find the upper bound of the ROUGE score achievable by the extractive summarization models.

We applied genetic algorithm (*Mitchell, 1998*), a nature-inspired technique used in many optimization problems applying the concepts of mutation and crossover. The algorithm is popular in the summarization models, both single and multi-document methods:

- Genetic algorithm application to maximize the fitness function, which mathematically expresses such summary properties as topic relation, readability, and cohesion (*Chatterjee, Mittal & Goyal, 2012*) in documents represented as a weighted directed acyclic graphs (DAG) (*Li & McCallum, 2006*) applying the popular graph methods in NLP (*Mihalcea & Radev, 2011*).
- The strength of genetic algorithms was demonstrated in finding optimal sentence feature weights for ETS methods. It was discovered that sentence location, proper noun,

**Table 1 Cleaned arXive dataset description.**

|        | Text length | Abstract length |
|--------|-------------|-----------------|
| count  | 17,038      |                 |
| mean   | 263.44      | 11.75           |
| std    | 102.57      | 2.13            |
| min    | 100.00      | 10.00           |
| 25%    | 179.00      | 10.00           |
| 50%    | 252.00      | 11.00           |
| 75%    | 338.00      | 13.00           |
| max    | 500.00      | 20.00           |

and named entity features get relatively higher weights because they are more critical for summary sentence selection (*Meena & Gopalani, 2015*).

- Vector representations produced by identifying and extracting the relationship between the input text main features and repetitive patterns, optimized by the genetic algorithm, used to generate precise, continuous, and consistent summaries (*Ebrahim et al., 2021*).

In the scope of our research, we are to apply a genetic algorithm to find the upper bound for summary quality achievable with the ETS methods. For example, *Simón, Ledeneva & García-Hernández (2018)* described a method based on a genetic algorithm to find the best sentence combinations of DUC01/DUC02 datasets in multi-document text summarization (MDS) through a meta-document representation.

# METHODS AND DATA

## Data

The arXive (arXiv.org) dataset, firstly introduced in 2018 (*Cohan et al., 2018*), contains 215K scientific articles in the English language from the astrophysics, math, and physics domains. The dataset comprises article texts, abstracts (reference or "golden" summary), article section lists, and main texts divided into sections.

Articles with abstracts that were accidentally longer than the main text and those with extremely long or short texts were excluded from the dataset. Thus, we end up with 17,038 articles with abstracts of 10 to 20 sentences; see Table 1.

## Methods

### Variable neighborhood search (VNS)

VNS is a metaheuristic method, exploiting the idea of gradual and systematical change in initial random solution space to find the approximate optimum of the objective function (*Burke & Graham, 2014*).

The VNS bases on the following facts (*Burke & Graham, 2014*):

1. Local minima of different neighborhood structures are not necessarily the same.
2. The global minimum is the same for all existing neighborhood structures.
3. In many problems, neighborhood structures local minima are close to each other.

---

*Initialization.* Select the set of neighborhood structures $\mathcal{N}_k$, for $k = 1, \ldots, k_{max}$, that will be used in the search; find an initial solution $x$; choose a stopping condition;
*Repeat* the following sequence until the stopping condition is met:
(1) Set $k \leftarrow 1$;
(2) *Repeat* the following steps until $k = k_{max}$:
    *(a) Shaking.* Generate a point $x'$ at random from the $k$th neighborhood of $x$ ($x' \in \mathcal{N}_k(x)$);
    *(b) Move or not.* If this point is better than the incumbent, move there ($x \leftarrow x'$), and continue the search with $\mathcal{N}_1$ ($k \leftarrow 1$); otherwise, set $k \leftarrow k + 1$;

---

**Figure 2  Pseudo-code for the reduced VNS.**

The pseudo-code of the reduced VNS, a variant of VNS that is not using the local search algorithm applied in this article, is given in Fig. 2.

### Greedy algorithm

A Greedy algorithm is any algorithm that follows the problem-solving heuristic of taking the best local solution for an optimization task (*Black, 2005*). For some problems, a greedy heuristic can yield locally optimal solutions approximating a globally optimal solution for a reasonable amount of time.

### Genetic algorithm

A genetic algorithm is a meta-heuristic method inspired by the natural process of selection belonging to the larger class of evolutionary algorithms. Genetic algorithms are widely used to generate solutions to optimization and search problems by using such operators as a crossover, mutation, and selection, which meet in adaptation and evolutionary processes of living species reproduction (*Mitchell, 1998*).

### Evaluation

We use recall-oriented understudy for gisting evaluation (ROUGE) scoring (*Lin, 2004*) for summary evaluation. The metric basic idea is in calculating the n-grams intersection percentage of reference (*recall*; see Eq. (1)) and candidate (*precision* summaries; see Eq. (2)). The harmonic mean integration between *recall* and *precision* is called the *F*1 score (Eq. (3)).

$$recall = \frac{len(R \cap C)}{len(R)}, \tag{1}$$

where $R$ and $C$ are the set of unique `n_grams` in reference and candidate summaries, and $len()$ is the number of words in a set.

$$precision = \frac{len(R \cap C)}{len(C)}. \tag{2}$$

$$F1\ score = 2 \times \frac{precision \times recall}{precision + recall}. \tag{3}$$

## EXPERIMENTS

In our previous article (*Akhmetov, Mladenović & Mussabayev, 2021b*) we searched for the best possible ROUGE-1 score using the VNS heuristic algorithm only. However, in this article, we added the ROUGE-2 score and applied greedy and genetic algorithms for comparison.

Using the Brute Force algorithm to find the combination of sentences yielding the highest ROUGE score has the $O(n!)$ computational complexity and therefore is not feasible; see Eq. (4). Therefore, we need to apply a heuristic algorithm to approximate the achievable upper level of summary quality.

We need to apply optimization algorithms because selecting the best possible combination of sentences for a summary from the original text using the Brute Force algorithm has the $O(n!)$ computational complexity and therefore is not feasible; see Eq. (4).

$$\begin{pmatrix} N_t \\ N_a \end{pmatrix} = \frac{N_t!}{N_a!(N_t - N_a)!} \qquad (4)$$

where $N_a$ and $N_t$ - are the respective number of sentences in summary and text.

Optimization algorithms provide a better alternative to Brute Force algorithms by generating not exact but an approximate and satisfactory solution using fewer computational resources and for a reasonable amount of time.

Therefore, we use VNS, greedy and genetic algorithms to find the best combinations of sentences from article texts yielding the highest ROUGE-1 score with original article abstracts as a reference.

### VNS

Using the VNS terminology, for every article in our dataset (Table 1), we cyclically applied the following procedures:

1. **Initial solution**: which is a randomly selected set of sentences $x$ in $\mathcal{N}_k = \begin{pmatrix} N_t \\ N_a \end{pmatrix}$ possible neighborhood structure space, for which we get the ROUGE-1 (*Lin, 2004*) score as the initial best solution to improve on.

2. **Shaking**: we change the initial solution by replacing a randomly selected sentence with a different one from the source text, increasing the rate of changes $k$ up to $k_{max}$ if no improvement in the ROUGE-1 score occurs, limiting the magnitude of the changes to a $k_{max}$ parameter ($k_{max} = 3$, three sentence replacements at a time in our case).

3. **Incumbent solution**: if the obtained summary ROUGE-1 score is better than the previous best solution, we fix the result and reset the $k$ to one sentence.

4. **Stop condition**: we limit the cycle by 60 s, 5,000 iterations, or 700 consecutive iterations without improvement of the ROUGE-1 score.

### Greedy algorithm

We used the following Greedy algorithm realization based on the general idea of the optimization algorithm of this class, where we try to find the most feasible immediate solution.

Given a source text (*T*) split into Sentences (*S*), and accompanied by its "golden" summary (*A*):

1. Compile a vocabulary of words from *A* as (*V*).
2. Create a word occurrence matrix (*M*), where we treat each item in *V* as columns, sentences in *T* as rows, and binary values indicating the presence of a word in a sentence.
3. Until matrix *M* is exhausted:

   - Sum the values in rows of *M* and get the maximum value sentence index, which is the index of the sentence containing the maximum number of words from the "golden" summary *A*. Store the obtained index in the Index List (*IL*).
   - Delete the columns in *M* for which the current maximum row values sum sentence has non-zero values.

4. To determine the optimal number of summary sentences for maximum ROUGE score:

   - Compute ROUGE score for every top-n sentences combination in *IL* ($1 \leq n \leq len(IL)$).
   - Select the *n* corresponding to the maximum ROUGE score.
   - Truncate *IL* to *n* top sentences.

5. To restore the initial sentence order in *T*, sort items in *IL* in the ascending order and assemble a summary by picking sentences from *T* with the respective indices in sorted *IL*.
6. Calculate the ROUGE score of the generated summary concerning *A*.

## VNS initialized by the Greedy

We worked on VNS initialized by the best results achieved by the Greedy algorithm. It is simply the modification of the algorithm described in "VNS" where we, instead of random initialization, use the sentences from the best summaries attained by the Greedy algorithm. Initialization of the VNS algorithm with a combination of sentences with a relatively high ROUGE score saves the time to achieve this initial result. Moreover, it sets the perspective to improve on top of the result achieved by a different algorithm.

## Genetic algorithm

Inspired by the results which evolutionary algorithms show in different applications (*Mitchell, 1998*), we developed a genetic algorithm realization for finding the upper bound for the ROUGE score.

Given a text (*T*) and its abstract (*A*):

1. Calculate lengths of *T* and *A* in number of sentences (*len_T* and *len_A*).
2. Shuffle the sentences in *T*.
3. Generate the initial generation of summary candidates by cutting the sentence list in T to chunks of the size *len_A*.
4. Set the number of offsprings to half the number of initial candidates (*n_offsprings*).

**Table 2 The best ROUGE scores (R-1 and R-2) achievable using ETS methods. Numbers in bold indicate the highest values by row.**

|       | VNS | | Greedy | | VNS_Greedy | | Genetic | | Genetic_Greedy | |
|-------|------|------|------|------|------|------|------|------|------|------|
|       | R-1 | R-2 | R-1 | R-2 | R-1 | R-2 | R-1 | R-2 | R-1 | R-2 |
| count | 17,038 | | | | | | | | | |
| mean | 0.55 | 0.21 | 0.55 | 0.23 | 0.58 | **0.25** | 0.58 | 0.24 | **0.59** | **0.25** |
| std | 0.07 | 0.08 | 0.08 | 0.10 | 0.08 | 0.10 | 0.07 | 0.09 | 0.08 | 0.10 |
| min | 0.07 | 0.01 | 0.04 | 0.01 | **0.09** | **0.02** | **0.09** | 0.01 | **0.09** | 0.01 |
| 25% | 0.52 | 0.16 | 0.51 | 0.16 | 0.54 | 0.18 | 0.55 | 0.18 | **0.56** | **0.19** |
| 50% | 0.56 | 0.20 | 0.55 | 0.21 | 0.58 | 0.22 | 0.59 | 0.23 | **0.60** | **0.24** |
| 75% | 0.59 | 0.25 | 0.60 | 0.28 | 0.62 | 0.29 | 0.63 | 0.29 | **0.64** | **0.30** |
| max | 0.84 | 0.78 | **0.97** | 0.93 | **0.97** | **0.95** | 0.86 | 0.84 | 0.92 | 0.88 |

5. Proceed for six generations:

(a) Crossover all candidates between each other by mixing the sentences of two candidates, shuffling them, and randomly selecting *len_A* number of sentences.
(b) Calculate the ROUGE-1 score for all the offspring.
(c) Select top *n_offsprings* by ROUGE-1 score and repeat.

6. Select the offspring from the last generation with the highest ROUGE-1 score and return it as the generated summary.

### Genetic algorithm initialized by the Greedy

This algorithm is the same as a randomly initialized "Genetic Algorithm". Nevertheless, in step 3, we add to the initial candidates the summary generated by the "Greedy Algorithm". The rationale behind initializing the Genetic algorithm with Greedy algorithm results is to improve on top of already high results, similar to the case in "VNS Initialized by the Greedy".

## RESULTS

Applying the the algorithms described in "Experiments" we show that the best results were achieved by the Genetic algorithm initialized by the results of Greedy algorithm 0.59/0.25 for the ROUGE-1/ROUGE-2 scores; see Table 2 and Fig. 3. While the best modern neural network models (*Zhang et al., 2019*; *Liu & Lapata, 2019*; *Lloret, Plaza & Aker, 2018*) can achieve ROUGE-1 of just 0̃.48 and ROUGE-2 of 0.22 on arXive dataset; see Table 3. So there is room for improvement in the ETS methods. Examples of the summaries produced by the algorithms employed in this article can be found at `https://github.com/iskander-akhmetov/Reaching-for-Upper-Bound-ROUGE-Score-of-Extractive-Summarization-Methods/blob/main/arXive_examples.md`.

Curiously, the maximum-ROUGE summaries from the five algorithms we used (VNS, greedy, genetic, VNS, and genetic initialized by greedy) differ in the average number of sentences: 15, 7, 12, 10, and 12, respectively. We attribute the reason that summaries

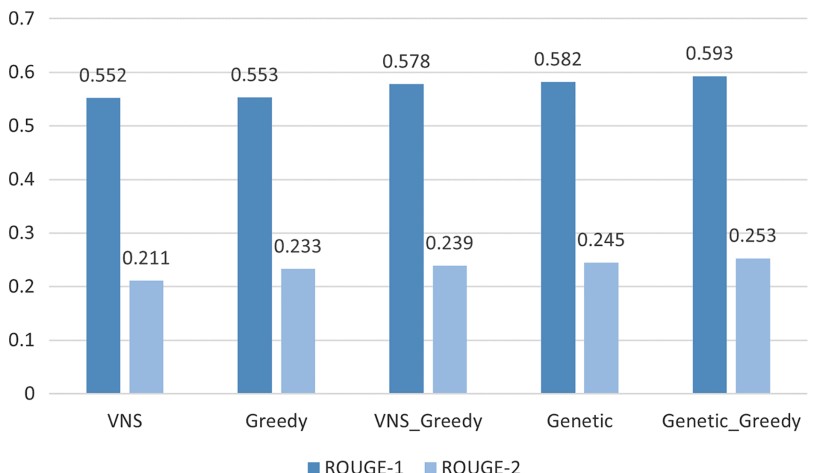

**Figure 3 Upper bound ROUGE scores comparison for different methods.**

**Table 3 Comparison of the upper bound obtained with the leading modern ATS models results on the arXive dataset. Numbers in bold indicate maximum values by column.**

| Class | Model | ROUGE-1 | ROUGE-2 |
|---|---|---|---|
| Genetic_Greedy upper bound | | **0.59** | **0.25** |
| Extractive | SumBasic (*Cohan et al., 2018*; *Lin, 2004*; *Vanderwende et al., 2007*) | 0.30 | 0.07 |
| | LexRank (*Cohan et al., 2018*; *Erkan & Radev, 2004*) | 0.34 | 0.11 |
| | LSA (*Cohan et al., 2018*; *Jezek, Steinberger & Ježek, 2004*) | 0.30 | 0.07 |
| Abstractive | Attn-Seq2Seq (*Cohan et al., 2018*; *Nallapati et al., 2016*) | 0.29 | 0.06 |
| | Pntr-Gen-Seq2Seq (*Cohan et al., 2018*; *See, Liu & Manning, 2017*) | 0.32 | 0.09 |
| | Discourse-att (*Cohan et al., 2018*) | 0.36 | 0.11 |
| | PEGASUSBASE (*Zhang et al., 2019*) | 0.35 | 0.10 |
| | PEGASUSLARGE (*Zhang et al., 2019*) | 0.45 | 0.17 |
| | BigBird-Pegasus (*Zaheer et al., 2020*) | 0.47 | 0.19 |
| | BertSumExtMulti (*Sotudeh, Cohan & Goharian, 2020*) | 0.48 | 0.19 |
| | LongT5 (*Guo et al., 2022*) | 0.48 | 0.22 |
| | PRIMERA (*Xiao et al., 2022*) | 0.48 | 0.21 |

generated by the Greedy algorithm have seven sentences on average to the fact that the algorithm purposefully chooses the lexically richest sentences, which are longer than average. The issue of selecting long sentences in favor of shorter ones was addressed in MMR paper (*Carbonell & Goldstein, 1998*), and the proposed solutions sought the balance between the relevance of the sentences and their length by weighing them according to the lexical unit's content. Conversely, VNS tries random sentence combinations not accounting for their properties. Thus, the Greedy algorithm maximizes the ROUGE score with fewer sentences than other algorithms. Moreover, determining the optimal number of sentences to maximize the summary ROUGE score is also challenging.

**Table 4 ROUGE1 error analysis.** Numbers in bold represent columns maxima.

| Algorithm | mean | std | CV | CI +/− mean | CI lower | CI upper |
|---|---|---|---|---|---|---|
| VNS | 0.5500 | 0.0700 | 0.1273 | 0.0011 | 0.5489 | 0.5511 |
| Greedy | 0.5500 | 0.0800 | 0.1455 | 0.0012 | 0.5488 | 0.5512 |
| VNS_Greedy | 0.5800 | 0.0800 | 0.1379 | 0.0012 | 0.5788 | 0.5812 |
| Genetic | 0.5800 | 0.0700 | **0.1207** | 0.0011 | 0.5789 | 0.5811 |
| Genetic_Greedy | **0.5900** | 0.0800 | 0.1356 | 0.0012 | **0.5888** | **0.5912** |

**Table 5 ROUGE2 error analysis.** Numbers in bold represent columns maxima.

| Algorithm | mean | std | CV | CI +/− mean | CI lower | CI upper |
|---|---|---|---|---|---|---|
| VNS | 0.2100 | 0.0800 | 0.3810 | 0.0012 | 0.2088 | 0.2112 |
| Greedy | 0.2300 | 0.1000 | 0.4348 | 0.0015 | 0.2285 | 0.2315 |
| VNS_Greedy | **0.2500** | 0.1000 | 0.4000 | 0.0015 | **0.2485** | **0.2515** |
| Genetic | 0.2400 | 0.0900 | **0.3750** | 0.0014 | 0.2386 | 0.2414 |
| Genetic_Greedy | **0.2500** | 0.1000 | 0.4000 | 0.0015 | **0.2485** | **0.2515** |

## Error analysis

We have performed an error analysis on the data obtained on ROUGE1/2 calculations for all of the methods employed in this research; see Tables 4 and 5. For each algorithm we have calculated the coefficient of variation (CV) defined in Eq. (5), and confidence interval (CI) defined in Eq. (6), with the confidence level of 95%.

$$CV = \frac{\sigma}{\mu}, \tag{5}$$

where $\sigma$ is the standard deviation (std) and $\mu$ is the mean.

$$CI = \mu \mp Z \times \frac{\sigma}{\sqrt{N}}, \tag{6}$$

where $Z$ is the Z-value associated with the desired confidence level (for 95% confidence level in our case, Z-score = 1.956), and $N$ is the number of observations.

We see in Table 4 that the genetic algorithm initialized by the Greedy algorithm results demonstrates the highest ROUGE1 score mean (0.5900) and highest values of upper (0.5912) and lower (0.5888) bounds of the CI. However, the genetic algorithm alone has the lowest CV (0.1207).

Table 5 shows that for ROUGE2 score means and CI upper and lower bounds are highest for both the VNS and genetic algorithms initialized by the results of the Greedy algorithm. Moreover, we see that CV values for the ROUGE2 score almost tripled, which means that these values are more dispersed and volatile than the ROUGE1 score average values. Moreover, again genetic algorithm has the lowest CV value.

11/16

## DISCUSSION

As we saw in our experiments, for ETS methods, selecting the optimal number of sentences to extract from the source text is detrimental to maximizing the ROUGE score of summaries. However, we detected no strong correlation between the optimal number of sentences for any of the algorithms and other factors such as the number of characters, words, and sentences in a source text and their derivative features (number of words per sentence or characters per word).

The summary length importance has been studied previously by *Jezek, Steinberger & Ježek (2004)*. However, they inferred by the latent semantic analysis (LSA) evaluation only that the more extended summaries are, the better. Their article was published the same year the ROUGE score was introduced by *Lin (2004)* to assess the summary quality automatically, which is now the summary evaluation "industry" standard. However, using the ROUGE score implies that more extended summaries increase the recall at the expense of precision. So further research is needed to determine the optimal number of summary sentences to maximize the ROUGE score value.

Another issue is that using the ROUGE scoring methodology presumes that the reference summaries are ground truth. However, we still have to check the "golden" summaries relative to their source text as they might be a teaser-style indicative summary. Alternatively, the reference summary we use in ROUGE scoring might be very abstractive, containing different wording than the source text, which leads ETS methods to failure.

A different question of whether the ROUGE metric suits the goal of measuring the information overlap of the generated summary with the golden summary was researched by *Deutsch & Roth (2021)*, and it was found that the metric instead measures the extent to which both summaries have the same topic. So there is a need to develop evaluation metrics to account for the informativeness of the generated summary relative to the source text and golden summary.

## CONCLUSION

We showed five algorithms to approximate the highest possible ROUGE score for ETS methods tested on the extract from the arXive dataset (*Cohan et al., 2018*). We used the VNS technique in our prior publication (*Akhmetov, Mladenović & Mussabayev, 2021b*), and in this article, we explored genetic and Greedy algorithms. The latter inspired us to develop a novel type of summarization algorithms (*Akhmetov, Gelbukh & Mussabayev, 2021a*). We showed that there is still a way to improve the ETS methods to reach the 0.59 ROUGE-1 score, while the latest contemporary summarization models do not surpass 0.48.

Our future work plan is to research:

1. Determine the optimal number of sentences in summary to maximize the ROUGE score in each case.
2. Narrowing the sentence search space for heuristic algorithms by excluding presumably unfit sentences (ex., too short sentences, and others).
3. Test the heuristic algorithms described here on different text summarization datasets.

## ACKNOWLEDGEMENTS

The authors thank the CONACYT for the computing resources brought to them through the Plataforma de Aprendizaje Profundo para Tecnologas del Lenguaje of the Laboratorio de Supercómputo of the INAOE, Mexico.

### Funding

This research is conducted within the Committee of Science of the Ministry of Education and Science of the Republic of Kazakhstan under the grant number AP09058174 in the course of "Development of language-independent unsupervised semantic analysis methods large amounts of text data" project. The work was done with the support from the Mexican Government through the grant A1-S-47854 of CONACYT, Mexico, and grants 20211784, 20211884, and 20211178 of the Secretaria de Investigación y Posgrado of the Instituto Politecnico Nacional, Mexico. The funders had no role in study design, data collection and analysis, decision to publish, or preparation of the manuscript.

### Grant Disclosures

The following grant information was disclosed by the authors:
Committee of Science of the Ministry of Education and Science of the Republic of Kazakhstan: AP09058174.
CONACYT, Mexico: A1-S-47854.
Secretaria de Investigación y Posgrado of the Instituto Politecnico Nacional, Mexico: 20211784, 20211884, and 20211178.

### Competing Interests

The authors declare that they have no competing interests.

### Author Contributions

- Iskander Akhmetov conceived and designed the experiments, performed the experiments, performed the computation work, prepared figures and/or tables, authored or reviewed drafts of the article, and approved the final draft.
- Rustam Mussabayev performed the experiments, analyzed the data, performed the computation work, prepared figures and/or tables, and approved the final draft.
- Alexander Gelbukh conceived and designed the experiments, analyzed the data, prepared figures and/or tables, authored or reviewed drafts of the article, and approved the final draft.

### Data Availability

The code is available at GitHub: https://github.com/iskander-akhmetov/Reaching-for-Upper-Bound-ROUGE-Score-of-Extractive-Summarization-Methods.

The data is available at Mendeley: Akhmetov, Iskander; Gelbukh, Alexander; Mladenovic, Nenad; Mussabayev, Rustam (2021), "The arXive dataset extract with

high ROUGE score summaries generated by 5 different methods", Mendeley Data, V1, DOI 10.17632/nvsxfcbzdk.1.

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
