# Peer review of "Reaching for upper bound ROUGE score of extractive summarization methods"

_PeerJ Computer Science, doi:10.7717/peerj-cs.1103_

## Round 0.1 · original submission · Major Revisions

I recommend a major revision to address the comments raised by the two reviewers. Please provide a detailed response letter. Thanks.

Reviewer 1 ·

Basic reporting

Lack of professional English and unclear. It is not clear what is the actual contribution of the authors.
Background section can be added to define existing methods used in the paper, in context with the paper.
Restructuring is required as some of the existing methods in the experimental section and in the section 3.
Results are ok but again explanation should be there, why and how are the parameters settings are done for the respective outcomes.

Formal results are clear under abstract.

The introduction section is too simple and doesn’t give a feeling of a research paper. More space of improvement.
The algorithms considered like VNS, greedy and genetic should be more explained in context of current paper.

Rephrase the last section in the Literature survey. The examples given by the authors should be written in a narrative form.

Grammar errors:
Much grammar improvement is required in the whole paper. Mentioning every error is quite a time and space-consuming.
off course
punctuation marks in the whole paper
Check for correct positions for capital letter usage.
“is not something new in ATS as we can bring as a few examples:” Rephrasing is needed

Experimental design

Avoid using the keywords like in the following phrase “In our previous article (Akhmetov et al., 2021b)”

How did the parameter tuning done by the authors?
Reasoning must be given for the variants of combinations of two algorithms used in the paper.
A little introduction to the comparison models should be given for understanding the purpose to the readers.
How reduced VNS is different from VNS.
Explanation of Tables and Figures used should be informative and more.
Paper doesn't seem to perform rigorous investigation.

Validity of the findings

The idea present in the paper is not novel and moreover, nothing is proposed, only existing methods are used.
No detailed description is given for variant of VNS algorithm.
Conclusion is a bit clear as compare to other sections of the paper.

Additional comments

Paper is not in a impressive readable form much improvement in terms of writing, detailed explanation, coherence of two pieces of texts is highly required.
Paper must not be accepted in the current form.

Reviewer 2 ·

Basic reporting

1. Literature references are not sufficient. Latest researches are not included except one or two. Need to include latest researches from past 2 years.
2. Sentences are long e.g. Line 147. Try to include shorter sentences.
3. There is a need to paraphrase many sentences as they are written more casually rather than technically. e.g. Paragraph above line 180. But there are many other instances.

Experimental design

1. Research is within Aims and Scope of the Journal.

2. Result Comparison with at least one more dataset is encouraged.

3. Inclusion of an example is encouraged including sentences and a paragraph using your technique.

Validity of the findings

1. The results are stated clearly but there is a need to assess the method on one more dataset.
2. Objective and Findings of the work are not clearly stated.
3. Error analysis is missing.

---

## Round 0.2 · Major Revisions

One reviewer recommended acceptance but there are some critical comments from the other reviewer. They should be addressed with a further revision.

Reviewer 1 ·

Basic reporting

I think the authors have worked on all the suggested reviews. After reading the paper, it seems that paper is clear and unambiguous now.

Experimental design

Improved

Validity of the findings

Improved

Reviewer 2 ·

Basic reporting

1. The way of writing the sub-points differ in Point 1, point 2 and point 3 on Page 1 and Page 2.
2. “The challenging question we want to answer in this paper is whether we have room for the ExtractiveText Summarization (ETS) methods development?” Page 2 Line 59. Incorrect structure.
3. Literature references are not sufficient. Latest researches are not included except one or two. Need to include latest researches from past 2 years. Previous answer is not satisfactory.
4. “We excluded from the dataset articles with abstracts accidentally longer than the original text”- Line 140, Line not clear
5. Many English sentences are not written clearly, paraphrasing is required at many places to make text more clear and understandable. Examples are quoted above also. Lack of professional English.

Experimental design

1. Labels on the graphs shown in Figure 3 are not visible and cannot be inferred.
2. “In this paper, we added the ROUGE-2 score and applied greedy and genetic algorithms for comparison.”- Line 172. How your work is novice?
3. Link to the code is missing.
4. Figure 3 is not explained, if it is providing any value addition or mere repetition of the facts shown in Table 2 and Table 3.

Validity of the findings

1. Unable to understand the novelty of this work. Please emphasize more on this.
2. Please mention your findings of the work
3. Still not clear what have you achieved in this work.

Additional comments

No

---

## Round 0.3 · accepted · Accept

The reviewers' comments have been addressed. The paper can be accepted.

Reviewer 2 ·

Basic reporting

I think the authors have worked on all the suggested reviews.

Experimental design

I think the authors have worked on all the suggested reviews.

Validity of the findings

I think the authors have worked on all the suggested reviews.

Additional comments

I think the authors have worked on all the suggested reviews.